# The Relation between Depressive Symptoms and Unsafe Sex among MSM Living with HIV

**DOI:** 10.3390/ijerph20021595

**Published:** 2023-01-16

**Authors:** Annemiek Schadé, Gerard van Grootheest, Johannes H. Smit

**Affiliations:** 1Department of Psychiatry, Amsterdam UMC, Vrije Universiteit Amsterdam, De Boelelaan 1117, 1081 HV Amsterdam, The Netherlands; 2Department of Psychiatry, Amsterdam UMC, Vrije Universiteit Amsterdam, Oldenaller 1, 1070 BB Amsterdam, The Netherlands

**Keywords:** HIV, depression, depressive symptoms, sex, unsafe sex, MSM

## Abstract

In people living with HIV (PLWH), a positive association is often found between depressive symptoms and unsafe sex, which means sex without a condom. However, the results of such studies are inconclusive. The present study compared the numbers of safe and unsafe sexual contacts from men who have sex with men (MSM) (N = 159), living with HIV and attending a mental health clinic, with those of HIV-negative MSM in the general population (N = 198). We determined whether the presence of depressive symptoms was associated with unsafe sex in either of the two study populations. The depressive symptoms were measured with the Inventory of Depressive Symptoms (IDS), (MSM living with HIV) and with the 2012 Sexual Health Monitor (HIV-negative MSM). Finally, we determined whether MSM living with HIV with depressive symptoms, who received psychiatric treatment as usual, engaged in fewer unsafe sexual contacts one year after baseline. The mental-health-treatment-seeking MSM living with HIV engaged in more unsafe sexual contact than the MSM comparison group without HIV. Neither the treatment-seeking MSM living with HIV nor the MSM without HIV in the general population exhibited a relationship between depressive symptoms and unsafe sex. Moreover, the successful treatment of depressive symptoms in the treatment group did not lead to any reduction in the number of unsafe sexual contacts. Further research is needed to develop interventions that might be effective for MSM living with HIV with mental health symptoms to reduce the number of unsafe sexual contacts.

## 1. Introduction

Mental health symptoms and mental disorders are common among people living with HIV (PLWH). In this group, the most common mental disorder is depression [1,2,3,4,5]. It is vital that mental health symptoms are recognised, diagnosed, and treated. This is because, in PLWH, these symptoms have an adverse impact on the patient’s adherence to treatment and on the prognosis of their HIV infection [6,7,8,9].

In PLWH, there is often a positive association between depression and depressive symptoms and unsafe sex. This mostly concerns depression, but there is also evidence for an association with drug use and borderline personality disorder [10,11,12,13,14,15]. It is a well-documented fact that untreated depression can increase the likelihood of unsafe sexual contact in PLWH [16,17,18,19]. Most of the research into the relation between mental health and unsafe sex in PLWH involved populations of MSM (men who have sex with men) [16,17,18,19]. It has been shown that psychological distress, such as depression, increases the likelihood that the affected individuals will engage in unsafe sex [17,19,20,21,22]. However, there is conflicting evidence in this regard, as other studies have found no such relation within HIV-positive MSM populations [17,23]. This discrepancy could reflect differences in study populations, in terms of the operationalisation of primary variables, and in the methodology used. Furthermore, some studies use the term ‘mental health’, while others use ‘depression’ or ‘depressive symptoms’. The operationalisation of sexual techniques, or the presence of an HIV-positive or HIV-negative partner, can also affect the research results [24,25]. In conclusion, most studies in this area have demonstrated the existence of a relationship between depression or depressive symptoms and unsafe sex in MSM living with HIV. However, this relation has not been universally confirmed by studies in this area, thus it remains an important subject for further research.

In the Netherlands, 27 hospitals are designated to diagnose and treat PLWH. Free HIV tests are available at public health institutions. Almost 60% of the PLWH in the Netherlands are MSM and these men represent 78% of the mental-health-treatment-seeking PLWH, in total 25,036 persons were registered with HIV [26]. In this population, depression and depressive symptoms are the most commonly cited mental health symptoms [8]. The prevalence of unsafe sex in the PLWH population is unknown. However, the majority of these men became infected through sexual contact [26], so it seems safe to assume that they are likely to have engaged in unsafe sexual contacts, which means sex without a condom. Unsafe sex can have significant repercussions for the patients themselves and for others, such as accidental infection with HIV, HCV (Hepatitis C virus) and other STDs (Sexually Transmitted Diseases). The results quoted in the literature are somewhat ambiguous with regard to the relation between depression and unsafe sex. This raises the question of whether treatment-seeking MSM living with HIV, with depression or depressive symptoms, engage in more unsafe sexual contacts than non-HIV-infected MSM.

Furthermore, it is important to know whether treating mental health symptoms, such as depression or depressive symptoms, has any impact on unsafe sexual contacts in PLWH. The literature contains very few studies on this subject. Some of these studies (which mainly involved PLWH with post-traumatic stress disorder) showed that the subjects engaged in fewer unsafe sexual contacts after receiving treatment for their mental health problems [27,28,29,30,31]. Sikkema et al. (2010) [27] proposed that mental health treatment can lead to substantial reductions in HIV transmission risk behaviour among patients living with HIV and that this should be a core component of secondary HIV prevention. No published studies have addressed the issue of whether treatment as usual for depression or depressive symptoms in MSM can lead to a reduction in unsafe sexual contacts. If treatment as usual is effective in reducing sexual risk behaviour, then this group of patients may require tailored interventions, to mitigate their risk of engaging in such contacts. Any further investigation of the above issues would require well-defined study populations.

The present study had three objectives. Firstly, we determine whether the number of safe and unsafe sexual contacts of MSM living with HIV, who were attending the HIV and mental health clinic, differed from that of HIV-negative MSM in the general population. Secondly, we determine whether the presence of depressive symptoms was associated with unsafe sex in either of the two study populations. Thirdly, we discover (by means of follow-up questionnaires) whether one year of psychological treatment as usual for MSM living with HIV, with depression or depressive symptoms, would lead to a reduction in unsafe sexual contacts.

## 2. Methods

Some of the MSM living with HIV attending a major HIV and mental health outpatient clinic in Amsterdam were found to have engaged in numerous sexual contacts, some of which were unprotected. We required a suitable comparison group, to determine whether there was a link between depressive symptoms and unprotected sex in this population. While we do have access to a cohort of HIV-negative MSM with depressive symptoms [8], we do not have any quantitative information concerning their sexual contacts. Fortunately, we also had access to a cohort of HIV-negative MSM without depressive symptoms that could serve as a comparison group. While it does feature some unusual characteristics, this is nevertheless a very suitable (and unique) comparison group. The questions on sexual contacts and mental health that we posed to the two study populations did differ in some respects. Furthermore, it would be preferable to investigate HIV and mental health independently. Nevertheless, these populations did enable us to obtain answers to our research questions.

In the present paper, we analysed data from two sources, a group of MSM living with HIV with a DSM-IV diagnosis and a group of MSM without HIV from the general population.

### 2.1. MSM from the General Population

MSM without HIV were selected from the 2012 Sexual Health Monitor. This was a representative online population study by the Rutgers WPF Knowledge Centre for Sexual and Reproductive Health and Rights, among 8064 subjects in the Netherlands [32]. This study included topics such as sexual contacts, condom use and sexually transmitted diseases (STDs). The study is a large-scale representative study of the sexual health of adults aged 18 to 80 years in the Netherlands (male/female) and included topics such as sexual contacts, condom use and sexually transmitted diseases (STDs). The study is part of a WHO initiative to monitor study sexual health in Latin America, Europe and Asia and Africa [33].

The following three questions (involving a six-point Likert scale ranging from ‘constantly’ to ‘never’) were used to indicate depressive feelings: “Did you feel so low that nothing could cheer you up?”, “Did you feel depressed and sad?”, and “Did you feel happy?”. The presence of depressive symptoms was defined as positive if the answer to one of the first two questions was ‘often’, ‘most of the time’, or ‘constantly’, or if the answer given in response to the third question was ‘seldom’ or ‘never’. We asked about the frequency of sexual contact, the number of sex partners in the past 6 months and the use of condoms [32].

For the purposes of our analyses, we selected a total of 198 subjects (HIV-negative men with a preference for MSM contacts) from the Rutgers Monitor. We did not measure a DSM diagnosis of depression, only depressive symptoms.

### 2.2. MSM from the HIV and Mental Health Cohort Study

MSM living with HIV were selected from our cohort study on HIV and Mental Health. Between 2006 and 2009, a total of 196 PLWH were included in this study cohort at GGZ InGeest in Geest’s (Amsterdam, The Netherlands) outpatient clinic for HIV and Mental Health [8]. This clinic specialises in the assessment and treatment of mental health problems, and in the use of alcohol and drugs among PLWH. At their first appointment, new patients were asked to participate in the study, and they gave written informed consent. The Medical Ethics Review Committee of the VU University Medical Center in Amsterdam approved the study.

At intake, DSM-IV diagnoses were obtained using the M.I.N.I. International Neuropsychiatric Interview (M.I.N.I. Plus 5.0) [34]. Self-reported questionnaires were administered at the first appointment and after a period of one year. These included the Inventory of Depressive Symptoms (IDS) [35]. We asked about the frequency of sexual contact, the number of sex partners in the past 6 months and the use of condoms [32].

For the present analyses, we selected MSM from the Mental Health cohort, resulting in a total of 159 MSM living with HIV. Once a diagnosis and treatment plan had been drawn up, all patients received either supportive therapy or cognitive behaviour therapy from a psychologist, psychiatrist, or a specialised mental health nurse. In some cases, this care as usual was supplemented with psychotropic medication. Unsafe sex was discussed during the course of treatment, but no specific interventions were used in this regard.

## 3. Analyses

The HIV-positive and the HIV-negative populations were compared by means of chi square tests and ANOVA; within-subject analyses were performed using Wilcoxon signed rank tests for dichotomous outcomes, and paired t-tests for continuous outcomes. All of the test data were analysed using IBM’s SPSS Statistics package.

## 4. Results

### 4.1. Comparison of MSM from the HIV and Mental Health Study with the General Population

The subjects in the HIV and Mental Health study did not differ significantly from those in the Rutgers Monitor, in terms of age (Rutgers: 43.3, HIV: 42.4; *p* = 0.466), partner status (Rutgers: 54%, HIV: 48%; *p* = 0.239), or employment (Rutgers: 60%, HIV: 66%; *p* = 0.296) (Table 1).

Compared to the Rutgers Monitor, the HIV study included a relatively larger number of subjects with a higher education qualification (Rutgers: 32%, HIV: 45%; *p* = 0.016). Both groups mostly consisted of people with a Western background (Rutgers: 92%, HIV: 87%; *p* = 0.100).

More subjects in the HIV and Mental Health cohort reported having sex with a new contact in the past 6 months than was the case in the Rutgers Monitor (Rutgers: 54%, HIV: 75%; *p* < 0.001). Moreover, the mean number of sex partners was higher among the HIV cohort than in the Rutgers Monitor (Rutgers: 4.1, HIV: 8.6 partners; *p* < 0.001). The HIV-positive group differed significantly from the general population with regard to unsafe sex. No fewer than 48% of the subjects in the HIV cohort (*p* < 0.001) reported having unprotected sex with a new contact in the past 6 months, whereas the corresponding figure for the Rutgers Monitor was just 20%.

### 4.2. Depressive Symptoms and Unsafe Sex within Both Study Populations

We assessed the two study populations for any association between unsafe sex and depressive symptoms or a diagnosis of depression (Table 2). Thirty-eight subjects (19%) from the Rutgers Monitor reported that they had experienced one or more depressive symptoms (‘often’, ‘most of the time’, or ‘constantly’) or that they were seldom or never happy. The presence of these symptoms was not related to the likelihood (*p* = 0.232) that these subjects had had sex in the past six months (see Table 2). With regard to those subjects who had had sex in the past six months, there was no association between their depressive symptoms and engaging in new sexual contacts (*p* = 0.104), nor was there a significant association between these symptoms and unprotected sex with new contacts (*p* = 0.156) or with the number of sexual partners in that six-month period (*p* = 0.720).

Similarly, in the HIV and Mental Health cohort, the presence of depression was not significantly associated with the likelihood that subjects had had sex in the past six months (*p* = 0.098). Within the group that did report having sex in the past six months, depression was not associated with the occurrence of new contacts (*p* = 0.552), with new unsafe contacts (*p* = 0.081), or with the number of sex partners in the past six months.

### 4.3. Changes in Unsafe Sex after One Year of Mental Health Care

Of the 159 MSM living with HIV and mental health problems, 72 subjects (45%) completed another questionnaire one year after intake. These patients showed a significant reduction in depressive symptoms, based on the IDS (from 31.1 (±12.8) points to 24.6 (±13.7) points; *p* < 0.001, Table 3). After one year, we observed no significant changes in the number of subjects who had had sex in the last six months (83% vs. 82%; *p* = 0.739). The number of subjects who reported having unsafe sex with new contacts before intake or after a period of one year remained stable (63% and 64%, respectively; *p* = 0.491), as did the number of sex partners over the past six months (10.5 ± 11.7 and 9.3 ± 12.5, respectively; *p* = 0.355).

## 5. Discussion

We obtained data on the number of safe and unsafe sexual contacts of MSM living with HIV who were attending the HIV and mental health clinic, to determine whether this group differed from the HIV-negative MSM in the general population in this regard. MSM living with HIV, with mental health symptoms, tended to engage in more unsafe sexual contacts than the MSM comparison group without HIV. As a result, they were previously at greater risk of becoming infected with HIV.

Several studies have described the relationship between depression, depressive symptoms and unsafe sex, especially in MSM populations living with HIV [17,19,20,21,22]. The cause of this relationship is multifactorial. The explanations given range from experiencing HIV-related stigma and self-efficacy for sexual safety, to childhood sexual abuse [16,17,18]. However, there is no universal agreement among studies in this area concerning the existence of a relationship between unsafe sexual contacts and depression and depressive symptoms in HIV-positive MSM [17,23]. In this study, we found no relationship between depression, depressive symptoms and unsafe sex in MSM living with HIV who were attending the HIV and mental health clinic and HIV-negative MSM in the general population.

We used follow-up questionnaires to determine whether psychological treatment as usual for MSM living with HIV with depression or depressive symptoms led to a reduction in the number of unsafe sexual contacts after one year. We found no evidence that the treatment of depressive symptoms resulted in a lower frequency of unprotected sexual encounters. The study’s power was limited by the level of non-response after one year. Nevertheless, we feel confident in asserting that the current dataset of 72 valid patient records shows no evidence that the frequency of unsafe sexual contacts was influenced by the treatment of depressive symptoms.

In both study groups, the presence of depressive symptoms was not related to the number of safe and unsafe sexual contacts. However, these groups differed slightly in the way in which depressive symptoms were assessed. Therefore, the MSM living with HIV does not completely match the general population group in this comparison. MSM living with HIV who were attending the HIV and mental health clinic have both HIV and depression or depressive symptoms. We are unable to disentangle the effects of these individual factors from the overall effect on sexual contacts. However, we do know that the MSM living with HIV with depression in our study group regularly engage in unsafe sex with a new partner. While depression or depressive symptoms did not increase the risk of unsafe sex, this risk may have been increased by other psychological factors not addressed in this study (such as borderline personality disorders) [12]. In addition, the substantial amount of drug abuse in our study group may have contributed to the incidence of unsafe sex [8].

Today, some members of the MSM living with HIV group are no longer infectious, due to their rigorous adherence to treatment with antiretroviral medication. However, other patients in this group are still at risk of transmitting or acquiring HIV, HCV and other STDs. From the point of view of public health, MSM living with HIV and PLWH in general with mental health disorders are a particularly interesting group. Not only are they already in care, but details of their risk behaviour are also known, which provides opportunities for the use of interventions with regard to sexual behaviour. In our outpatient treatment clinic for HIV and Mental Health, the matter of unsafe sexual contacts is routinely discussed. In this case, this measure does not seem to have been sufficiently effective or specific. Currently, there are no specific interventions designed to reduce the number of unsafe sexual contacts in PLWH with mental health disorders. Even though PrEP (pre-exposure profylaxis) is now available, not all men will use it and it is not available everywhere. Furthermore, several non-mental health factors are associated with unsafe sexual contacts. Nevertheless, there are opportunities to develop such an intervention programme in the context of mental health care [36]. Further research is needed to develop interventions specifically designed to reduce unsafe sex in PLWH with depression or other mental health symptoms [37].

## 6. Limitations of the Study

The two study groups differed in the way in which depression or depressive symptoms were assessed and sexual contacts measured, which could have influenced the results. Ideally, we would like to have used a comparison group of depressive MSM without HIV, but no such cohort was available. We are explicitly not comparing the two populations in terms of their mental health, because the instruments cannot be compared. However, in Table 1, we describe the sociodemographic characteristics of the two study groups, as well as the occurrence of sexual contacts.

This questionnaire-based study involved a high level of non-response. For this reason, we have refrained from making any claims about the effectiveness of treatment. Nevertheless, the dataset does allow us to answer the question of whether those responders whose depressive symptoms showed some improvement also showed any changes in terms of the numbers engaging in unprotected sex with new contacts. We concluded that the frequency of such contacts after treatment was the same as it had been before.

## 7. Conclusions

We found no relationship between depression or depressive symptoms and unsafe sex in MSM living with HIV who were attending the HIV and mental health clinic and HIV-negative MSM in the general population. However, MSM living with HIV and PLWH in general with depressive symptoms or other mental health symptoms regularly engaged in unsafe sex with new partners. There are no specific interventions designed to reduce the number of unsafe sexual contacts in PLWH with mental health symptoms. Further research is needed to develop interventions specifically aimed at reducing unsafe sex in PLWH with depressive or other mental health symptoms.

## Figures and Tables

**Table 1 ijerph-20-01595-t001:** Comparison between MSM from the general population (Rutgers Sexual Health Monitor) and from the study on HIV and Mental Health.

	Rutgers Sexual Health Monitor (HIV −) (N = 198)		HIV and Mental Health Cohort (HIV +) (N = 159)		*p*-Value
Mean age	43.3 (±14.5)		42.4 (±8.3)		0.466
Has a partner	107	54%	71	48%	0.239
Has paid work	119	60%	99	66%	0.296
Higher education	64	32%	65	45%	0.016
Western background	182	92%	135	87%	0.100
Ever tested for HIV	97	65%	-	-	-
Has had sex in the past 6 months	173	87%	134	84%	0.402
sex with a new contact	106	54%	119	75%	<0.001
unsafe sex with a new contact	39	20%	76	48%	<0.001
mean no. of sex partners	4.1 (±9.5)		8.6 (±10.1)		<0.001

**Table 2 ijerph-20-01595-t002:** Comparison of sexual contacts between subjects with and without depressive symptoms or diagnosis performed separately within the two study populations.

Rutgers Sexual Health Monitor	Depressive Symptoms					
	No. (N = 160)		Yes (N = 38)		*p*-Value	
Has had sex in the past 6 months	142	89%	31	82%	0.232	
sex with a new contact	91	64%	15	48%	0.104	(within the group that had sex)
unsafe sex with a new contact	35	25%	4	13%	0.156	(within the group that had sex)
no. of sex partners	4.3 (±9.7)		3.6 (±8.9)		0.720	(within the group that had sex)
**HIV and Mental Health Cohort**	**Depression Diagnosis**					
	**No (N = 75)**		**Yes (N = 84)**		* **p** * **-Value**	
Has had sex in the past 6 months	67	89%	67	80%	0.098	
sex with a new contact	58	88%	61	91%	0.552	(within the group that had sex)
unsafe sex with a new contact	33	49%	43	64%	0.081	(within the group that had sex)
no. of sex partners	9.2 (±11.7)		8.0 (±8.4)		0.466	(within the group that had sex)

**Table 3 ijerph-20-01595-t003:** Safe and unsafe sexual contacts at intake and after one year.

N = 72 HIV and Mental Health Cohort	at Intake		after One Year		*p*-Value	
IDS score	31.1	±12.8	24.6	±13.7	<0.001	
Has had sex in the past 6 months	60	83%	59	82%	0.739	
sex with a new contact	56	93%	51	88%	0.102	(within the group that had sex at T0 or T1)
unsafe sex with a new contact	38	63%	41	64%	0.491	(within the group that had sex at T0 or T1)
no. of sex partners	10.5	(±11.7)	9.3	(±12.5)	0.355	(within the group that had sex at T0 or T1)

## Data Availability

The data presented in this study are available on request from the corresponding author. The data are not publicly available due to eg privacy or ethical.

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
