# Peer review of "The Relation between Depressive Symptoms and Unsafe Sex among MSM Living with HIV"

_ijerph, 2023, doi:10.3390/ijerph20021595_

Round 1

Reviewer 1 Report (Previous Reviewer 1)

The relation between depressive symptoms and unsafe sex among MSM living with HIV

Thank you for the opportunity to review this important and interesting article. This is a previously revised manuscript, which allows for previous corrections to have been implemented, therefore, this is an optimized version of the paper. Hence, I believe that the article has now sufficient quality to be published.

Best wishes.

Reviewer 2 Report (Previous Reviewer 4)

The authors have carried out all the corrections mentioned by the reviewers.

This manuscript is a resubmission of an earlier submission. The following is a list of the peer review reports and author responses from that submission.

Round 1

Reviewer 1 Report

The relation between depressive symptoms and unsafe sex among MSM living with HIV

Thank you for the opportunity to review this article. This is an interesting and important contribution to the field. Overall, the article is well written and methodologically strong. Still, I believe a few changes would improve the quality of the manuscript and would increase its chances of being published:

1.     Abstract: please include information regarding the methods in the abstract, namely, number of participants, measurement instruments and procedures. Also, please adhere to the journal’s format.

2.     Lines 63-88: authors fundamentally describe the study and their methodological choices. This should be in a different topic other than the introduction (“The present study”).

3.     Authors would be welcome to include one or two paragraphs regarding the positive or protective factors when explaining lines 42-50.

4.     Authors should include some information regarding the HIV situation in the Netherlands, including epidemiological information, access to health services (including barriers) and cultural contexts.

5.     Line 89 – please replace the verbs “we hoped to”, and “we wanted to know”, with objective and clear statements of what your objectives were, to provide more clarity and objectivity.

6.     Why was DSM-IV used and not DSM-5? Please explain in the text.

7.     Data are 10 years old. Please provide rationale as to this is still updated to the present time, since so much can happen over a decade (covid-19, mental health policies in your country, antiretroviral medication, etc.).

8.     Authors only say that “this was a representative online population study”. This must be thoroughly explained: please provide more information on the gathering of the data. Also, if this was a convenience sample collected online, randomization wasn’t applied, therefore, it is not representative.

9.     Instruments measures must be clearly described in the methods section. Don’t forget to include the assessment of unsafe sex behavior. What was asked?

10.  Results. Please include a table with detailed sociodemographic information.

11.  Table 2 does not reflect any association. Please rewrite and explain properly. Authors compared two groups.

12.  Discussion: The major flaw of this study is the comparison made between the clinical sample using the DSM and the non-clinical sample using 3 questions that do not constitute a diagnosis, only symptoms. I believe this is not adequate do be done.

13.  Authors should include an implications section.

Best wishes.

Author Response

  1. Abstract: please include information regarding the methods in the abstract, namely, number of participants, measurement instruments and procedures. Also, please adhere to the journal’s format.

We added in the abstract the number of participants and the following sentences:

The depressive symptoms were measured with the Inventory of Depressive Symptoms (IDS), (MSM living with HIV) and with the 2012 Sexual Health Monitor (HIV-negative MSM)

  1. Lines 63-88: authors fundamentally describe the study and their methodological choices. This should be in a different topic other than the introduction (“The present study”).

We thank the reviewer for pointing out that a part of the introduction is actually a description or operationalization of our research question and we agree that it should be a part of the methods section. We start the method section now with this paragraph:

Some of the MSM living with HIV attending a major HIV and mental health outpatient clinic in Amsterdam were found to have engaged in numerous sexual contacts, some of which were unprotected. We required a suitable comparison group, to determine whether there was a link between depressive symptoms and unprotected sex in this population. While we do have access to a cohort of HIV-negative MSM with depressive symptoms [8], we do not have any quantitative information concerning their sexual contacts. Fortunately, we also had access to a cohort of HIV-negative MSM without depressive symptoms that could serve as a comparison group. While it does feature some unusual characteristics, this is nevertheless a very suitable (and unique) comparison group. The questions on sexual contacts and mental health that we posed to the two study populations did differ in some respects. Furthermore, it would be preferable to investigate HIV and mental health independently. Nevertheless, these populations did enable us to obtain answers to our research questions.

  1. Authors would be welcome to include one or two paragraphs regarding the positive or protective factors when explaining lines 42-50.

We understand that we are pointing out some specific (negative challenges). But we feel that introducing protective factors will lead to a side line in the paper which also need context and references. In this case we decided not to change it, also because the other reviewers did not mention this as needed.

  1. Authors should include some information regarding the HIV situation in the Netherlands, including epidemiological information, access to health services (including barriers) and cultural contexts.

The following text is added: In the Netherlands, 27 hospitals are designated to diagnose and treat PLWH. Free HIV-tests are available at the public health institutions. Almost 60% of the PLWH in the Netherlands are MSM and these men represent 78% of the mental-health-treatment-seeking PLWH, in total 25,036 persons were registered with HIV [22

  1. Line 89 – please replace the verbs “we hoped to”, and “we wanted to know”, with objective and clear statements of what your objectives were, to provide more clarity and objectivity.

We changed the text: Firstly, we determine whether the number of safe and unsafe sexual contacts of MSM living with HIV, who were attending the HIV and mental health clinic, differed from that of HIV-negative MSM in the general population. Secondly, we determine whether the presence of depressive symptoms was associated with unsafe sex in either of the two study populations. Thirdly, we discover (by means of follow-up questionnaires) whether one year of psychological treatment as usual for MSM living with HIV, with depression or depressive symptoms, would lead to a reduction in unsafe sexual contacts.

  1. Why was DSM-IV used and not DSM-5? Please explain in the text.

During our research period, the DSM-5 was not available yet

  1. Data are 10 years old. Please provide rationale as to this is still updated to the present time, since so much can happen over a decade (covid-19, mental health policies in your country, antiretroviral medication, etc.).

Ten years later, unsafe sexual contacts are still an important issue. Early treatment of the HIV-infection with medication is more common, but still many MSM are at risk of HIV of other SDT’s. The mental health policy in the Netherlands has not changed. Covis-19 reduced temporarily the possibility of (anonymous) sexual contact, but at the moment the situation is comparable to the pre-covid-19 situation.

  1. Authors only say that “this was a representative online population study”. This must be thoroughly explained: please provide more information on the gathering of the data. Also, if this was a convenience sample collected online, randomization wasn’t applied, therefore, it is not representative.

We added the following text:

The study is a large-scale representative study of the sexual health of adults aged 18 to 80 years in the Netherlands (male / female) included topics such as sexual contacts, condom use, and sexually transmitted diseases (STDs). The study is part of a WHO initiative to monitor study sexual health in Latin America, Europe and Asia and Africa [37].

  1. Instruments measures must be clearly described in the methods section. Don’t forget to include the assessment of unsafe sex behavior. What was asked?

We added in the method section the following sentences: We asked about frequency of sexual contact, number of sex partners in de past 6 months and use of condoms.

  1. Results. Please include a table with detailed sociodemographic information.

Only demographics shown are present in both studies

  1. Table 2 does not reflect any association. Please rewrite and explain properly. Authors compared two groups.

We have changed the title of table 2:

Table 2. Comparison of sexual contacts between subjects with and without depressive symptoms or diagnosis, performed separately within the two study populations.

  1. Discussion: The major flaw of this study is the comparison made between the clinical sample using the DSM and the non-clinical sample using 3 questions that do not constitute a diagnosis, only symptoms. I believe this is not adequate do be done.

We added the following sentences in the limitation section:

We are explicitly not comparing the two populations in terms of their mental health, because the instruments cannot be compared. However in table 1 we describe sociodemographic characteristics of the two study groups, as well as the occurrence of sexual contacts.

  1. Authors should include an implications section.

In the discussion, the following lines are intended as implication:

Today, some members of the MSM living with HIV group are no longer infectious, due to their rigorous adherence to treatment with antiretroviral medication. However, other patients in this group are still at risk of transmitting or acquiring HIV, HCV, and other STDs. From the point of view of public health, MSM living with HIV and PLWH in general with mental health disorders are a particularly interesting group. Not only are they already in care, but details of their risk behaviour are also known, which provides opportunities for the use of interventions with regard to sexual behaviour. In our outpatient treatment clinic for HIV and Mental Health, the matter of unsafe sexual contacts is routinely discussed. In this case, this measure does not seem to have been sufficiently effective or specific. Currently there are no specific interventions designed to reduce the number of unsafe sexual contacts in PLWH with mental health disorders. Even though PREP is now available, not all men will use it and it is not available everywhere. Furthermore, several non-mental-health factors are associated with unsafe sexual contacts. Nevertheless, there are opportunities to develop such an intervention programme in the context of mental health care [31].

Reviewer 2 Report

Thank you for the opportunity of reading the article. I have some remarks to it:

The issue and the subject of the article is interesting and needed as there is a lack of such theme in the current literature and researches. The biggest doubt of the scientific soundness and quality of the article is the date of presenting researches. The analysed data are outdated (2006-2009-2012) and not relevant to the current status of the knowledge in the field. Since 2012 there are significant changes of the quality of life of MSM living with HIV/AIDS thanks to effective treatment, PrEP, many prevention campaigns, positive social attitudes regarding MSM and PLWH. Moreover due to the ARV treatment U=U (undetectable = untransmittable) PLWH can have unprotected sexual contacts that are safe for their partners so it can not be stated that lack of using condoms is the matter of unsafe behaviours. The discussion lacks clarity on above issues. It is also unclear whether depressive symptoms were present in both study groups. It also would be more appropriate and scientifically interested to compare the population of MSM HIV+ having depressive symptoms and MSM HIV- having these symptoms than the general population of MSM and MSM HIV+ with depressive symptoms. In addition, the references list contains most of the items from 5-10 years ago, whose data and analyses are no longer up to date. References to current and recent literature are needed.

Author Response

We agree with the reviewer that a comparison between the population of MSM HIV+ having depressive symptoms and MSM HIV- having these symptoms could be a scientifically interesting topic. However we do not have this opportunity given the available data.

In the introduction, 4 recent references are added.

In the discussion, we mention PREP and the fact that most of PLWH are no longer infectious:

Today, some members of the MSM living with HIV group are no longer infectious, due to their rigorous adherence to treatment with antiretroviral medication. However, other patients in this group are still at risk of transmitting or acquiring HIV, HCV, and other STDs. From the point of view of public health, MSM living with HIV and PLWH in general with mental health disorders are a particularly interesting group. Not only are they already in care, but details of their risk behaviour are also known, which provides opportunities for the use of interventions with regard to sexual behaviour. In our outpatient treatment clinic for HIV and Mental Health, the matter of unsafe sexual contacts is routinely discussed. In this case, this measure does not seem to have been sufficiently effective or specific. Currently there are no specific interventions designed to reduce the number of unsafe sexual contacts in PLWH with mental health disorders. Even though PREP is now available, not all men will use it and it is not available everywhere.

Reviewer 3 Report

-References should be updated. Most of the are more than 10 years old.

Major concerns are:

-The validity of the measurement of depressive symptoms in the Rutgers sample is doubtful, at its best. I think it should be excluded from this study.

-Low comparability between MH cohort study and Rutgers sample. Moreover, I think comparability should be further explored with other relevant variables.

-No evidence about the quality of the measurements (validity, reliability) has been provided for the MH cohort study.

-No clue in the manuscript about what test is used to test each objective.

-No effect size and confidence interval has been reported.

-The lack of comparability between groups demands an alternative statistical test to Anova.

Author Response

-The validity of the measurement of depressive symptoms in the Rutgers sample is doubtful, at its best. I think it should be excluded from this study.

-Low comparability between MH cohort study and Rutgers sample. Moreover, I think comparability should be further explored with other relevant variables.

The Rutgers sample is not ideal in measuring depressive symptoms, but it is the best sample available and we did measure depressive symptoms. Therefore, we do not exclude it from the study.

In the limitation section, we added some extra lines:

The two study groups differed in the way in which depression or depressive symptoms were assessed and sexual contacts measured, which could have influenced the results. Ideally, we would like to have used a comparison group of depressive MSM without HIV, but no such cohort was available. We are explicitly not comparing the two populations in terms of their mental health, because the instruments cannot be compared. However win table 1 we describe sociodemographic characteristics of the two study groups, as well as the occurrence of sexual contacts.

-No evidence about the quality of the measurements (validity, reliability) has been provided for the MH cohort study.

-No clue in the manuscript about what test is used to test each objective.

It is written in the method section, line 137

-No effect size and confidence interval has been reported.

-The lack of comparability between groups demands an alternative statistical test to Anova.

The reviewer gives no indication on what alternative test to use. The other reviewers did not mention this in their comments. The purpose of the study, the compromises made by finding a study to compare the psychiatric cohort with a population cohort to compare sexual behaviours of MSM made us decide not to follow this advice.

As we are comparing means we think that an Anova with a p-value is correct and easily understood by the readers and confidence interval and effect size will give the readers a wrong sense of precision. See: Morey, R.D., Hoekstra, R., Rouder, J.N. et al. The fallacy of placing confidence in confidence intervals. Psychon Bull Rev 23, 103–123 (2016). https://doi.org/10.3758/s13423-015-0947-8

Reviewer 4 Report

The authors have designed and implemented a valuable study. The article also was well-written. All the concerns raised were explained in the discussion and and in the limitations.

I have just a comment. Give abbreviation for PREP in line 241, for easy understanding for the readers.

Author Response

We added: PrEP (pre exposure profylaxis)